# Digital Forensic Analysis to Improve User Privacy on Android

**DOI:** 10.3390/s22113971

**Published:** 2022-05-24

**Authors:** Hyungchan Kim, Yeonghun Shin, Sungbum Kim, Wooyeon Jo, Minju Kim, Taeshik Shon

**Affiliations:** 1Platform Tech Team, WINS Co., Ltd., Seongnam 13487, Korea; hj1003hj@ajou.ac.kr; 2Department of Artificial Intelligence Convergence Network, Ajou University, Suwon 16499, Korea; syh2347@ajou.ac.kr (Y.S.); zx1962@ajou.ac.kr (S.K.); klklkl098@ajou.ac.kr (M.K.); 3Department of Computer Engineering, Ajou University, Suwon 16499, Korea; dndusdndus12@gmail.com; 4Department of Cyber Security, Ajou University, Suwon 16499, Korea

**Keywords:** user privacy, permanent deletion, digital forensic, Android forensic, Android filesystem, Ext4 filesystem

## Abstract

The Android platform accounts for 85% of the global smartphone operating-system market share, and recently, it has also been installed on Internet-of-Things (IoT) devices such as wearable devices and vehicles. These Android-based devices store various personal information such as user IDs, addresses, and payment information and device usage data when providing convenient functions to users. Insufficient security for the management and deletion of data stored in the device can lead to various cyber security threats such as personal information leakage and identity theft. Therefore, research on the protection of personal information stored in the device is very important. However, there is a limitation that the current research for protection of personal information on the existing Android platform was only conducted on Android platform 6 or lower. In this paper, we analyze the deleted data remaining on the device and the possibility of recovery to improve user privacy for smartphones using Android platforms 9 and 10. The deleted data analysis is performed based on three data deletion scenarios: data deletion using the app’s own function, data deletion using the system app’s data and cache deletion function, and uninstallation of installed apps. It demonstrates the potential user privacy problems that can occur when using Android platforms 9 and 10 due to the leakage of recovered data. It also highlights the need for improving the security of personal user information by erasing the traces of deleted data that remain in the journal area and directory entry area of the filesystem used in Android platforms 9 and 10.

## 1. Introduction

The Android platform is a smartphone operating system based on the Linux kernel. It accounted for 83% of the global Android smartphone operating-system market share in 2021. Recently, with the development of information and network technology, the Android platform has been expanded and used in IoT devices such as wearable devices, vehicles, and artificial intelligence (AI) speakers. These IoT devices generate a large amount of data that are stored either in the cloud or the device. As sensitive information such as a user’s personal information and activities are also stored in the cloud and on the device there exists a potential risk of information leakage [1].

For example, there is a steady increase in cases wherein the personal information of previous users is sold by recovering deleted data from used Android smartphones [2,3]. In addition, it is necessary to protect personal information stored on IoT devices because secondary damage such as illegal logins to other sites, thefts of payment names, and financial damage may occur as a result of leaked personal information. In particular, the general smartphone-replacement cycle is short—less than two years—and new smartphones are equipped with Android platforms 9 and 10 as standard; therefore, further study on improving personal information security is needed [4].

With the emergence of this global need for personal-information protection, the General Data Protection Regulation in the EU, California’s Consumer Privacy Act, and the Stop Hacks and Improve Electronic Data Security Act in the USA have been implemented. Accordingly, a method of permanently deleting and managing data to ensure personal-information protection in IoT devices using the Android platform is needed.

However, Android-related studies published thus far have been focused on the collection and analysis of data stored in Android devices, and the extant studies related to personal-information protection are thus insufficient [5,6,7,8,9].

In addition, the Android platform can change the storage structure, encryption, and processing of deleted data through periodic version updates. When the data storage structure and deletion data processing method changes, previously nonexistent data may remain on the device, or previously remaining data may not remain on the device. For this reason, it may be difficult to apply existing studies performed on Android. Therefore, as the latest operating system is installed and released on smartphones and continuous updates are provided, research related to personal information protection for the latest Android platform should be conducted. The contributions of this paper are as follows:Through a filesystem forensic study on Android platforms 9 and 10, it was confirmed through experiments that deleted data can be recovered through the journal area and unassigned area of the filesystem when a user deletes data artificially.An experiment to analyze deleted data and recoverability was conducted for the system default application and a user-installed messenger application used in Android smartphones. The experiment was conducted by proposing three user data deletion scenarios: the app self-deletion function, the system app data and cache wipe function, and the installed application uninstallation function. The data deletion scenario was set based on the data deletion method mainly used in actual smartphones.Among the proposed three deletion verification scenarios, it was confirmed that data recovery using the journal area of the database file after data deletion using the application’s own function is possible. It was confirmed that the file name and file binary remain after the TRIM function.

The remainder of this paper is structured as follows: Section 2 deals with the existing studies related to mobile forensics, filesystems, and permanently deleted Android platform data. Section 3 deals with the filesystem metadata analysis of the Android platform related to user privacy, and Section 4 verifies the filesystem metadata changes and possibility of recovery before and after file deletion on the Android platform based on three deletion scenarios. Section 5 deals with traces of deleted data remaining on Android smartphones and the potential privacy problems that may arise, as well as limitations. Section 6 discusses the results of this study and future studies.

## 2. Related Studies

Research on the Android platform is actively being conducted in areas such as acquiring and analyzing data stored in smartphones, permanently deleting data, and analyzing filesystems [10,11,12,13,14,15,16,17,18,19,20]. In 2018, a study was conducted to acquire and analyze banking and public transportation application data stored on Android smartphones [7]. It was determined that the user’s personal information and the user’s actions are stored in the smartphone. In 2019, five collection and analysis methods were proposed for domestic AI speakers using the Android platform [21]. It was found that user-related data are transmitted and received while communicating with the cloud and stored in the AI speaker and the connected smartphone.

Research to derive the recoverability of deleted data on Ext4 filesystems has been continuously conducted since 2012 [22,23,24]. Through these studies, it was derived that the metadata of changed data are backed-up in the journal area when data are changed in the filesystem, and it was shown that recovery of deleted data is possible. However, these studies did not analyze the Ext4 filesystem installed in flash memory in the Android environment and did not analyze metadata that changed after data deletion.

In 2017, a study was conducted to identify defects in the Android smartphone data deletion mechanism by analyzing data deletion, uninstalled applications, and factory reset algorithms from Android smartphones. It was shown that important data can be recovered even after data are deleted in the case of Galaxy S3, Nexus 4, and Nexus 7 devices under Android 6.0, and the need for security enhancement was emphasized [9].

The majority of the existing studies have been focused on the collection and analysis of data stored in Android devices. In addition, research conducted on devices using Android platforms 9 and 10 is insufficient, and Android’s filesystem and the protection of the stored data in the device are not considered.

## 3. Analysis of Android Filesystem Metadata for User Data Deletion and Recovery

### 3.1. Inode and Directory in Android Ext4 Filesystem Metadata

In the metadata structure of the Ext4 filesystem, inode stores basic file information. Fields directly related to file recovery in the inode structure are shown in Table 1. When user data are deleted on the Android platform, the deletion time, size, and extent tree in the inode structure, may change. The extent tree is important from a data recovery perspective because it represents the area where the binary of the file is stored. A deleted file can be recovered by finding the area where the file’s binary is stored, through the extent tree, and extracting as much as the size field of the file. Therefore, in terms of user privacy, the file size and extent tree fields should be deleted from the inode after data deletion.

In the metadata structure of the Ext4 filesystem, directory entries store the name of the file in the directory, the size of the directory, and the inode number. Fields directly related to file recovery in directory entries are shown in Table 2. When user data are deleted on the Android platform, inode and name fields may be deleted from directory entries. The inode number is important from a recovery perspective because it is used to find the address of the inode where information about the file is stored. The deleted files can be recovered through inode analysis after finding the inode address through the inode number field in directory entries. Additionally, the name of the deleted file can be recovered through the name field of directory entries. Therefore, in terms of privacy, the name and inode number fields should be deleted from the directory entries after data deletion.

### 3.2. Journaling Space Analysis on Android and Linux

A journal superblock, journal descriptor block, and journal data block exist in the journaling space of the Ext4 filesystem. The journal superblock stores the basic information of the journaling space. The journal descriptor block stores the journal block tag array indicating the location of the journal data block. The journal data block stores inode backups, and when files are created or deleted, inodes are changed and then backed-up in the journal area.

The size of the initial journal area is determined by the number of blocks, and the number of blocks determines the size of the partition. If the partition size is greater than 64 GB and less than 128 GB, the journal area is allocated 256 MB, and if the partition size is greater than 256 GB, the journal area is allocated 1 GB. Further, in the case of the Samsung Galaxy S9+, which supports the Android platform, the size of the initial journal area does not change even if the device usage increases.

A comparison of the journal area of Android 9 and 10 for the Samsung Galaxy S9+ and the journal area of a PC is shown in Table 3. The detailed structure of the journal area on Android platforms 9 and 10 and PC differs depending on the journal checksum version. The journal block tag structure depends on the journal checksum version but does not affect the recovery of deleted data. The size of the journal area affects recovery because the maximum number of inodes that can be backed-up changes. For user privacy, even the journal data block of the journaling space, where the inode is backed-up after data deletion, must be completely deleted.

TRIM means that the operating system deletes data in blocks that are no longer used in the filesystem. Blocks that are not used in the filesystem are called “unallocated areas”. In general, unallocated areas that occur when a file is deleted from the filesystem include data area, directory entry, and inode. The data area refers to an area that stores the binary of actual data, and the directory entry and inode refers to an area that stores information about directories and files. For smartphones that support the Android platform, if the TRIM function is not used, only the link to the location where the binary of the data is located is removed when data are deleted; thus, data recovery is possible. Whether the TRIM function of an Android smartphone can be used or not can be checked through the fstab file. The fstab file of the Galaxy S9+ is shown in Figure 1, and the TRIM function is enabled by default. Even if the TRIM function is activated, data recovery is possible if the operation cycle of TRIM is long, or the area related to deleted file recovery is not changed to an unallocated area.

## 4. Scenario-Based Verification Related to Privacy Issues on Android Platform

In this section, we analyze the filesystem before and after user data deletion using Android platforms 9 and 10. Afterward, the possibility of recovering deleted personal information for each deletion scenario and application is determined.

### 4.1. Overview of Experiment Scenarios

Smartphones that support the Android platform store data, such as personal information and smartphone usage data, when providing various services to users. If the user’s personal information stored on the device is not properly managed, it may be exposed to security threats such as personal information leakage and identity theft. To identify security threats that can occur in real smartphones supporting Android platforms 9 and 10, three data deletion scenarios and analysis target applications were selected. In addition, the possibility of recovering deleted personal information is experimentally confirmed based on the deletion scenario and application. The data deletion scenarios—used in real smartphones that support Android platforms—for personal information recovery verification experiments are shown in Table 4. Scenario I is a method to delete application data using the application’s own function. When deleting data, users can select and delete only the data they want to delete as the most used method. Scenario II is a method to delete data by using the data and cache wipe function of the system setting menu. This method can delete application data in bulk. Scenario III is a method of uninstalling installed applications with the uninstall command in the system setting menu. This method is used when the installed application is no longer in use.

The applications selected for the analysis of the three previously described data deletion scenarios are shown in Table 5. The analysis target applications were selected as frequently used system default applications and user-installed messenger applications to emphasize the risks of personal information leakage and name theft that could result from the recovery of deleted data. Message App and Gallery App were selected as system default applications, and Facebook Messenger App and KakaoTalk App were selected as user-installed messenger applications. The KakaoTalk App has the highest market share of mobile apps in Korea. It is a messenger application that enables messages to be sent and received between smartphone users. The smartphone used for the verification experiment is a Galaxy S9+, which supports Android platforms 9 and 10. Detailed information of the smartphone is shown in Table 6. The smartphone used in this experiment was Samsung Electronics’ Android smartphone selected by referring to the global market share in 2021 [25].

Prerequisites for the privacy verification experiment of Galaxy S9+ targets are shown in Table 7. The experiment can be performed through filesystem analysis after acquiring the userdata partition. Administrator privilege is required to acquire the userdata partition. It is necessary to acquire administrator privileges because the Galaxy S9+ is released with user privileges. Additionally, in the Galaxy S9+ using Android platform 10, the userdata partition is set to Full Disk Encryption by default, and thus, it is necessary to release the encryption function. In this experiment, administrator privilege was obtained through the Odin program to meet the above prerequisites. Afterward, the full disk encryption function was disabled by modifying the forceencrypt flag of the fstab file that stores the partition’s filesystem information. Disabling the encryption of userdata partitions was only applied to the Samsung Galaxy S9+ using Android Platform 10. The userdata partition image in the unencrypted state can be acquired because the flag in forceencrypt changed from encryptable to footer and encryption is disabled.

### 4.2. Scenario I: Experiment with Delete Function Provide by the Applications

#### 4.2.1. System Default App: Message App

Android’s Message app provides a function to send and receive SMS/MMS with another person. The data generated during its use are stored in a database file and include personal information, such as phone numbers and conversation details. For the experiment, conversation contents with and without the keyword “delete” were created. After that, only the conversation contents containing the keyword “delete” were deleted, and the Scenario I experiment was conducted. The comparison of the mmssms.db file inode before and after the deletion is shown in Figure 2. The size of the file and the modification time of the database has changed, but the extent representing the data area has not. If the mmssms.db file is extracted based on the inode after deleting the conversation contents of the message, the deleted conversation contents cannot be identified as shown in Figure 3a. Furthermore, even when journal-area-based data recovery is performed, the deleted data cannot be recovered because the backed-up inode points to the same data area.

The mmsms.db file is a required file that is initially created when an application is installed, and it is not deleted. Instead, only the data area of the mmsms.db file is deleted. However, when using the data recovery is possible through the database journal file because write ahead logging (WAL) is set by default. The message application’s database journal file is created as mmssms.db-wal. The deleted conversation remains in the mmssms.db-wal file as shown in Figure 3b and can be restored.

#### 4.2.2. System Default App: Gallery App

Android’s Gallery app provides a function to view photographs taken by the user, downloaded from the Internet, and received through messenger applications. The photographs are saved as general files with the .jpg and .png extensions, and in the case of photographs taken directly, location information is also included in the photograph’s metadata. The Scenario I experiment was conducted by deleting photographs downloaded from the Internet. The comparison of the photograph file inode before and after the deletion is shown in Figure 4a. The file size and extent indicating the data area for the photograph file are all changed to 0x00. As the extent field is changed to 0x00 after data deletion, the data area, where the binary of the photograph file is stored, cannot be found. The deleted data cannot be recovered based on the inode. However, if the deleted photograph is extracted based on the backed-up inode in the journal area, the original photograph can be obtained. This is because the field values of the inode before data deletion and the backed-up inode in the journal area after deletion are the same, as shown in Figure 4b. Due to the Android filesystem using the journal area, deleted data can be recovered based on the backed-up inode in the journal area. In particular, even in an environment wherein TRIM operates on Android, data recovery is possible if TRIM has not been performed after the data deletion. This is because when Android deletes data, only the extent indicating the data area where the data binary is stored is deleted, and the deleted data binary remains as an unallocated area.

#### 4.2.3. User Installed App: Facebook Messenger App

Android’s Facebook Messenger app provides a function to send and receive SMS/MMS with another person. The data generated during its use are stored in a database file and include personal information such as phone numbers and conversation details. In the experiment using Scenario I, content containing keyword “delete” was deleted from the conversation. Similar to when data are deleted in the Message app, only the modification time of the database file is changed. As shown in Figure 5a, if the threads_db2 file is extracted based on the inode after the conversation is deleted, the deleted data cannot be checked; thus, the same result as the Message app can be obtained. However, there are cases where the Facebook Messenger app can recover data through the db-journal file that stores the database file log even after data deletion, as shown in Figure 5b.

#### 4.2.4. User Installed App: KakaoTalk App

Android’s KakaoTalk app is a messenger application that provides the function to send and receive SMS/MMS with another person. The data generated during its use are stored in the kakaotalk.db file, including personal information such as phone number, username, e-mail, and profile image. The KakaoTalk app supports “delete for me” and “delete for everyone” as data deletion methods. In this experiment, the analysis was performed after deleting the data using two methods supported by the KakaoTalk app. Using Scenario I, content containing keyword “delete” was deleted from the conversation. The modification time and size change the same as when data are deleted from the Message app. The result of extracting the kakaotalk.db file based on the inode after deleting the conversation is shown in Figure 6. If the data were deleted using the delete-for-me method, the deleted conversation could not be confirmed, but if the data were deleted using the delete-for-everyone method, the deleted conversation can be confirmed. As the conversation contents of the KakaoTalk app are encrypted and stored in a database file, the conversation contents were inferred based on the created_at field and the deleted_at field. Due to the backed-up inode in the journal area also pointing to the same data area, it is impossible to recover data even with backed-up inodes in the journal area.

Through the verification experiment on the possibility of recovering deleted data using Scenario I, it was confirmed that the Message app, Gallery app, Facebook Messenger app, and KakaoTalk app can recover data even after the app data are deleted. In the case of a messenger app where data are stored in a database file, recovery was possible through the journal file of the database. In the case of the Gallery app, where data are stored in a general file, recovery was possible through the journal area. Therefore, even if the app data are deleted using Scenario I, traces of the deleted data remain, and user privacy problems may occur.

### 4.3. Scenario II: Experiment with the System Data & Cache Wipe

#### 4.3.1. System Default App: Message App

The Message app usage data are stored in the “/user_de/0/com.android.providers.telephony/” directory. The Message app data are not deleted because data deletion using the data and cache wipe function of the Android smartphone system settings deletes the app data stored in the “/data/data/” subdirectory.

#### 4.3.2. System Default App: Gallery App

The Gallery app usage data are stored in the “/media/0/” directory. The Gallery app’s data are not stored in the “/data/data/” subdirectory, and thus they are not deleted the same as the Message app’s results.

#### 4.3.3. User Installed App: Facebook Messenger App

The Facebook Messenger app usage data are stored in “/data/data/com.facebook.orca/database/threads_db2”. The comparison of inodes in the “/data/data/com.facebook.orca/databases” directory before and after data deletion using Scenario II is shown in Figure 7a. After data deletion, the file size, modification time, and extent are all changed. The data cannot be recovered based on inode because the extent representing the data area of the database directory, where threads_db2 is stored, is changed to 0x00.

In the case of database directory backed-up inodes in the journal area, all fields are the same as for database directory inodes before data are deleted, as shown in Figure 7b. However, if the database directory entry is found based on the backed-up inode in the journal area, as shown in Figure 7c, the binary of the data cannot be determined, thus the data cannot be recovered based on the journal area.

#### 4.3.4. User Installed App: KakaoTalk App

The actual usage data of the KakaoTalk application are stored in “/data/data/com.kakao.talk/database/kakaotalk.db”. The comparison of inodes in the “/data/data/com.kakao.talk” directory before and after data deletion using Scenario II is shown in Figure 8a. After data deletion, the inode number of the database directory is changed, and it can be observed that the inode number of the database directory, before the deletion, is allocated as the inode number of the no_backup directory. As the inode number has been changed, it is impossible to trace the upper directory where kakaotalk.db is stored, and inode based data recovery is not possible. The comparison of the kakaotalk.db backed-up inode in the journal area and the kakaotalk.db inode before data deletion is shown in Figure 8b, and all fields are identical. However, if the kakaotalk.db file is found based on the backed-up inode in the journal area, it cannot be recovered based on the journal area because the binary file of the data cannot be checked as shown in Figure 8c.

### 4.4. Scenario III: Experiment with the Application Uninstall

The experimental results of Scenario III applied to Message, Gallery, and Facebook Messenger apps are almost identical to those of Scenario II. However, the experimental results of applying Scenario III to the KakaoTalk app are different from the experimental results of Scenario II. Therefore, in this section, the experimental results are mainly described with a focus on the differences from Scenario II.

#### 4.4.1. System Default App: Message App

The “uninstall applications” function of smartphones that support Android platforms is only available for user-installed applications. Therefore, in the case of the Message app installed by default, Scenario III cannot be used, and data recovery cannot be determined.

#### 4.4.2. System Default App: Gallery App

In the case of the Gallery app, which is installed by default, the possibility of data recovery cannot be determined because Scenario III cannot be performed as with the Message app.

#### 4.4.3. User Installed App: Facebook Messenger App

Experimental results of Scenario III applied to Facebook Messenger app are the same as Scenario II, data recovery based on inode and journal area cannot be performed.

#### 4.4.4. User Installed App: KakaoTalk App

The comparison of filesystem metadata before and after data deletion of KakaoTalk app using Scenario III is shown in Figure 9. As the extent indicating the data area for the kakatalk.db file is changed to 0x00, inode based data recovery cannot be performed, as in the results of Scenario I’s Gallery app. The data area where the binary of the deleted data is found, based on the backed-up inode stored in the journal area, is shown in Figure 9d. Due to the binaries of the deleted data remaining in the data area, the deleted data can be recovered based on the journal area.

### 4.5. Privacy Issues According to Data Deletion Scenarios

The results of the recovery possibility verification experiment of data after data deletion using three data deletion scenarios are shown in Table 8. When data is deleted using Scenario I, data recovery is possible in the Message app, Gallery app, Facebook Messenger app, and KakaoTalk app. If data are stored in a database file such as Message app, Facebook Messenger app, or KakaoTalk app, they can be recovered through the db-wal or db-journal file. When data are stored as a general file as in the Gallery app, they can be restored based on the journal area. When data are deleted using Scenario II, data recovery is possible for both the Message and Gallery app, but data recovery is impossible for the Facebook Messenger app and KakaoTalk app. As the app data of system defaults apps such as the Message app and Gallery app are not stored in the “/data/data/” subdirectory, data are not deleted, and data recovery is possible based on inode and journal area. User-installed messenger apps, such as Facebook messenger and KakaoTalk, cannot find the database file where app data are stored, and thus data recovery based on inode and journal area cannot be performed. When data are deleted using Scenario III, data recovery is possible in the KakaoTalk app, but data recovery is impossible in the Facebook Messenger app. In the case of the Message app and Gallery app, Scenario III cannot be applied; therefore, it is not possible to determine whether data recovery is possible. In the KakaoTalk app, all fields of the backed-up inode in the journal area and the inode before deletion are the same, and data can be recovered through the journal area. In the Facebook Messenger app, all fields of the backed-up inode in the journal area and the inode before deletion are the same, but the data area where the binaries of data are stored is deleted, and thus data recovery is impossible. As mentioned above, it was confirmed that there are cases where application data are not deleted, even after deletion, and there are cases where recovery is possible for smartphones that support the Android platforms 9 and 10.

## 5. Discussion

We conducted an experiment on the possibility of recovering deleted data for two types of system default applications and two types of user-installed messenger applications for Samsung Galaxy S9+, using Android platforms 9 and 10. In these experiments, there is a precondition that rooting must be performed to obtain administrator privileges and disable encryption functions. These constraints may make it difficult to apply erasure data recovery methods, as verified by the experiments detailed in this paper, to general smartphones, but they are significant as an attempt to improve user privacy.

An experiment on recoverability was conducted using three data deletion methods: the self-deletion function of the app, the data and cache wipe function of the system app, and the uninstallation of the installed application. Scenario I is a data deletion method using the app’s own deletion function. For Scenario I, it was confirmed that even after data deletion, recovery is possible based on the journal file of the database file and the backed-up inode in the filesystem journal area. Scenario II is a data deletion method using the data and cache wipe function of the system app. In the case of Scenario II, it was confirmed that no data deletion target occurred after data deletion or information about the file name remained in directory entries. In particular, it was confirmed that the directory entries area is not deleted even if the TRIM function of Android operates. Scenario III is a method of deleting data by uninstalling installed applications. For Scenario III, it was confirmed that data recovery is possible based on the backed-up inode in the journal area after data deletion. In addition, it is expected that it will be helpful to employ the method of erasing the traces of data left in the journal area of the Android filesystem or the unallocated area of the directory entry to improve the privacy protection of Android smartphones.

## 6. Conclusions

An analysis performed to determine whether traces of deleted data remain on Android smartphones is an important research topic from the perspective of user privacy. In this paper, we analyzed the filesystem metadata before and after user data deletion, according to the deletion scenario for smartphones supporting Android platforms 9 and 10 and determined the possibility of data recovery. In particular, we confirmed that data recovery is possible in the Messages, Gallery, KakaoTalk, and Facebook Messenger applications even after deleting the data using the applications’ own function, which is predominantly used by users. It was confirmed that cyber security threats, such as personal information leakage and identity theft, may occur through a verification experiment. In addition, the importance and necessity of data management and permanent deletion measures to improve personal information protection were verified. Therefore, future research should focus on methods of improving personal information protection that can be applied to actual smartphones, supporting the Android platform. In addition, research should be conducted to improve user privacy in devices such as AI speakers and smart watches that use the Android platform.

## Figures and Tables

**Figure 1 sensors-22-03971-f001:**
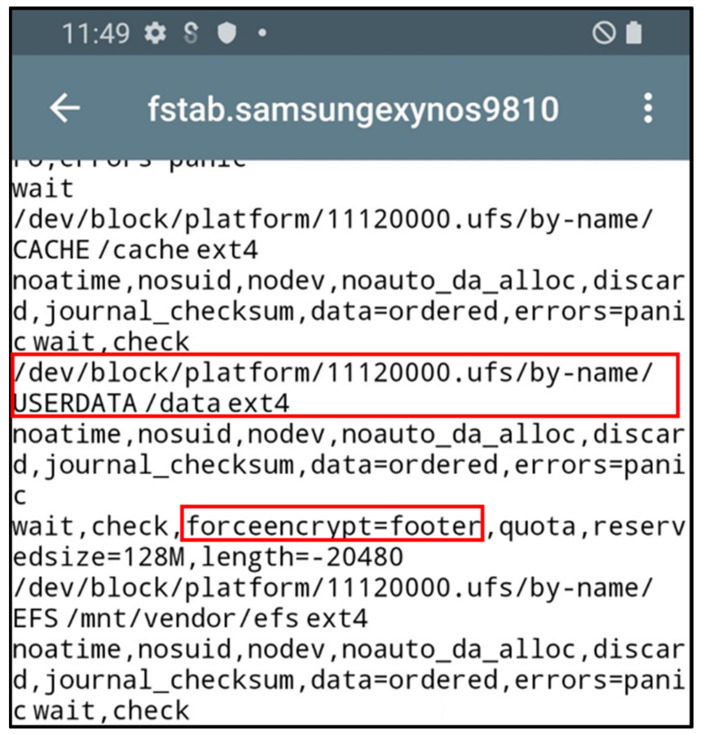
Galaxy S9+ TRIM function default value in fstab.samsungexynos9810 file.

**Figure 2 sensors-22-03971-f002:**
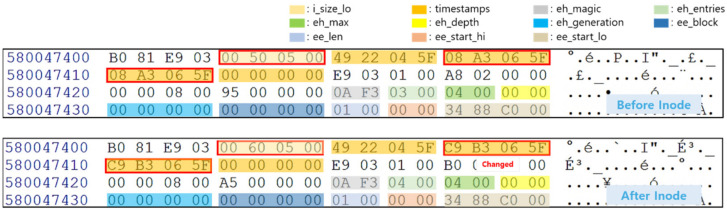
Comparison of mmssms.db’s inode before and after deleting app data using Scenario I. The red box indicates changed fields before and after deletion.

**Figure 3 sensors-22-03971-f003:**
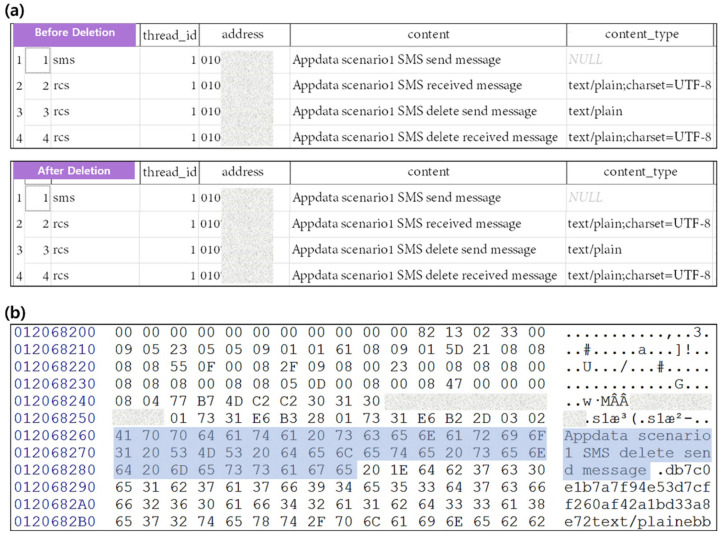
Analysis of db file and db-wal file before and after Message data deletion using scenario I (**a**) Comparison of mmssms.db before and after app data deletion (**b**) Traces of deleted data remaining in mmssms.db. The blue box indicates data remaining after deletion.

**Figure 4 sensors-22-03971-f004:**
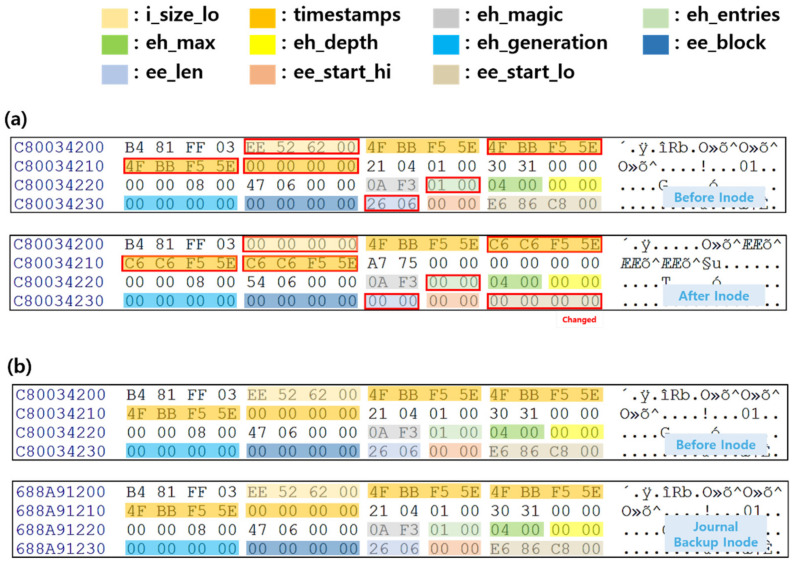
Metadata analysis of filesystem before and after app data deletion using scenario I (**a**) Comparison of image1.jpg’s inode before and after deleting app data (**b**) Comparison of image1.jpg’s inode before deletion and journal backup inode the name of the journal file of the database. The red box indicates changed fields before and after deletion.

**Figure 5 sensors-22-03971-f005:**
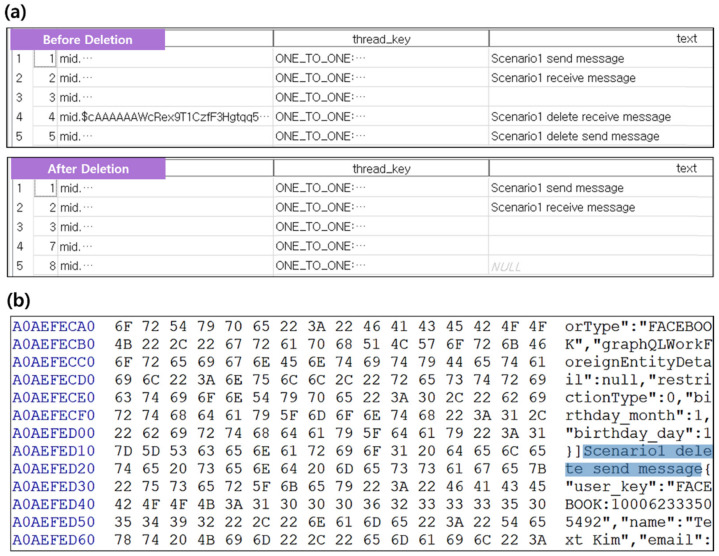
Analysis of db files and db-wal files before and after data deletion using scenario I (**a**) Comparison of threads_db2’s before and after app data deletion (**b**) Traces of deleted data remaining in threads_db-journal.

**Figure 6 sensors-22-03971-f006:**
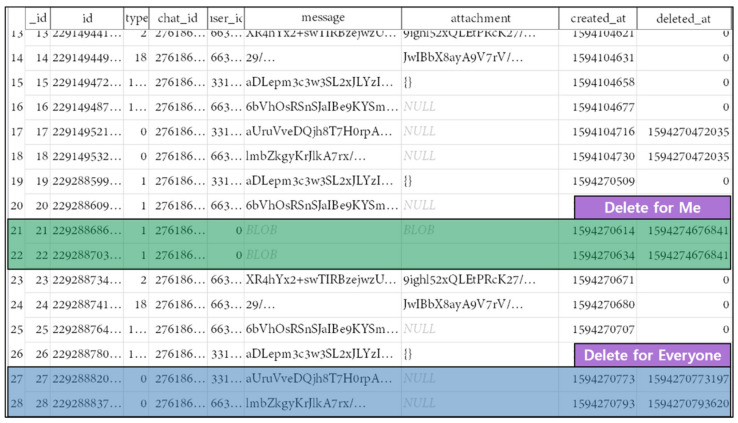
Inode based kakaotalk.db recovery after deleting app data using Scenario I.

**Figure 7 sensors-22-03971-f007:**
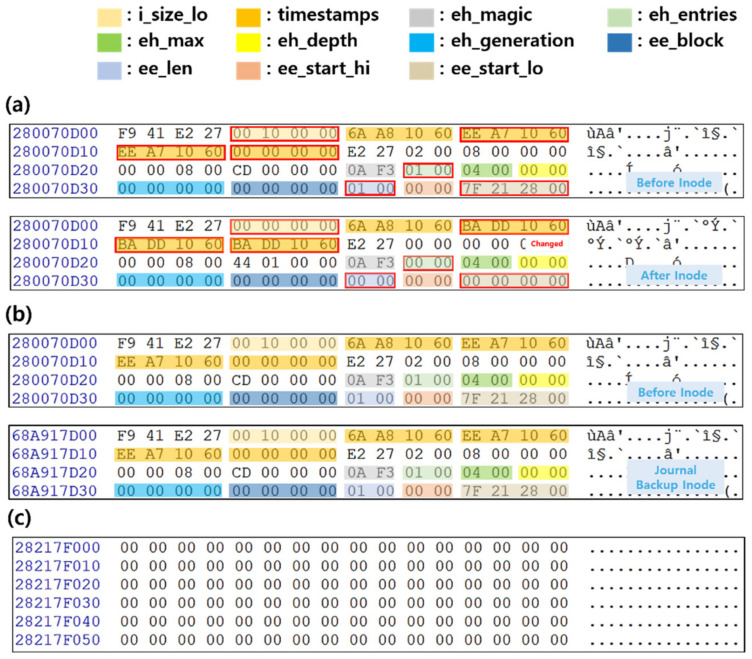
Metadata analysis of filesystem before and after Facebook Messenger data deletion using scenario II (**a**) Comparison of /data/data/com.facebook.orca/database’s inode before and after deleting app data (**b**) Comparison of /data/data/com.facebook.orca/databases’s inode before deletion and journal backup inode (**c**) Database directory entries searched based on journal backup inode.

**Figure 8 sensors-22-03971-f008:**
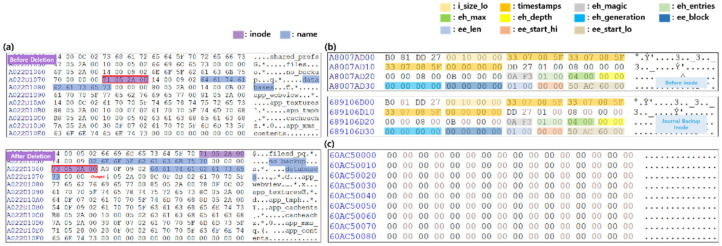
Metadata analysis of filesystem before and after data deletion using scenario II (**a**) Comparison of/data/data/com.kakao.talk’s inode before and after deleting app data (**b**) Comparison of kakaotalk.db’s inode before deletion and journal backup inode (**c**) Kakaotalk.db’s data area searched based on journal backup. The red box indicates changed fields before and after deletion.

**Figure 9 sensors-22-03971-f009:**
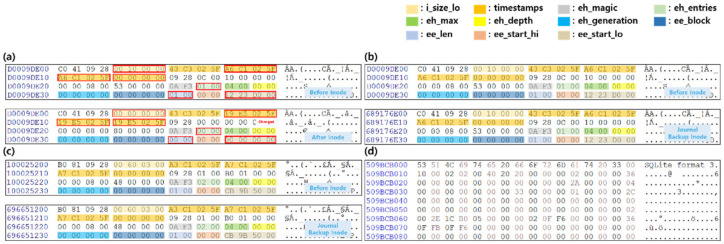
Metadata analysis of filesystem before and after data deletion using scenario III (**a**) Comparison of/data/data/com.kakao.talk’s directory entries before and after deleting app data (**b**) Comparison of/data/data/com.kakao.talk’s inode before deletion and journal backup inode (**c**) Comparison of kakaotalk.db’s inode before deletion and journal backup inode (**d**) Comparison of kakaotalk.db’s data area found based on journal backup inode. The red box indicates changed fields before and after deletion.

**Table 1 sensors-22-03971-t001:** Main structure of inode related to file recovery.

Offset	Size (Byte)	Filed Name	Description
0x04	4 Byte	i_size_lo	Lower 32-bits of size in bytes
0x08	4 Byte	i_atime	Last access time, in seconds since the epoch
0x0C	4 Byte	i_ctime	Last inode change time, in seconds since the epoch
0x10	4 Byte	i_mtime	Last data modification time, in seconds since the epoch
0x14	4 Byte	i_dtime	Deletion Time, in seconds since the epoch
0x28	60 Byte	i_block	Extent tree (Data area address related value)

**Table 2 sensors-22-03971-t002:** Main structure of directory entries related to file recovery.

Offset	Size (Byte)	Filed Name	Description
0x00	4 Byte	inode	Number of the inode that this directory entry points to
0x04	4 Byte	rec_len	Length of this directory entry
0x06	4 Byte	name_len	Length of the file name
0x07	1 Byte	file_type	File type code
0x08	N Byte	name	File name

**Table 3 sensors-22-03971-t003:** Comparison of Android and PC journal size.

	Ubuntu 20.04 1 TB	Galaxy S9+ 64 GB		Galaxy S9+ 256 GB	
Android Version	-	9	10	9	10
Linux Kernel	5.4.0	4.9.59	4.9.118	4.9.59	4.9.118
Journal Checksum	v3	v2	v2	v2	v2
Default Journal	1 GB	256 MB	256 MB	1 GB	1 GB

**Table 4 sensors-22-03971-t004:** Android application data deletion scenarios.

Data Deletion Scenario	Description	Target Application
Scenario I	Deleting data using the application’s own function	Message, Gallery, Facebook Messenger, KakaoTalk
Scenario II	Deleting data using the data and cache wipe of system applications	Message, Gallery, Facebook Messenger, KakaoTalk
Scenario III	Deleting data using the application uninstall *	Facebook Messenger, KakaoTalk

* System default application cannot be uninstalled.

**Table 5 sensors-22-03971-t005:** Android application used in experiment.

Application Name	Version	Description
Message App	11.5.10.406	System default application
Gallery App	11.5.02.6	
Facebook Messenger App	297.0.0.6.119	User installed application
KakaoTalk App	9.1.8	

**Table 6 sensors-22-03971-t006:** Android device used in experiment.

Device (Storage)	Version (Kernel)
Android Galaxy S9+ (64 GB)	Android 9 (Kernel 4.9.5)
	Android 10 (Kernel 4.9.115)
Android Galaxy S9+ (256 GB)	Android 9 (Kernel 4.9.5)
	Android 10 (Kernel 4.9.115)

**Table 7 sensors-22-03971-t007:** Experimental requirements to improve user privacy.

Experiment Device (Version)	Requirements
Android Galaxy S9+ (64 GB)	Administrator Authority
Android Galaxy S9+ (256 GB)	Administrator Authority
	Userdata Partition Decryption

**Table 8 sensors-22-03971-t008:** Possibility of application data recovery according to data deletion scenarios.

Data Deletion Scenario	Target Application	Data Recovery
Scenario I	Message App	Using db-wal file
	Gallery App	Using journal area
	Facebook Messenger	Using db-journal file
	KakaoTalk	Using db-wal file
Scenario II	Message App	Not deleted
	Gallery App	
	Facebook Messenger	Unrecoverable
	KakaoTalk	
Scenario III	Facebook Messenger	Unrecoverable
	KakaoTalk	Using journal area

## Data Availability

Not applicable.

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
