# Peer review of "Digital Forensic Analysis to Improve User Privacy on Android"

_sensors, 2022, doi:10.3390/s22113971_

Round 1

Reviewer 1 Report

The topic of this study is very interesting. The manuscript is well written. However, in order to improve the quality of the manuscript, my suggestions are given below.

1) In the introduction section please mention the benefits of this research, highlight your contribution using bullets or points and also add the motivation of this research study.

2) Improve the quality of figures, i.e. figures 3

3) Please include the organization of the paper at the end of the introduction section. For example; Overall, this study is organized as follows. 

4) Please add related work to this study. Also, add more recent studies in this section. In addition, please add a comparison table and compare the most recent studies along with advantages and disadvantages.

5)In performance evaluation you should mention the simulation tool along with details.

6)In the conclusion section please mention the future research of your study.

Author Response

A reply letter is attached as an attachment.

Reviewer 2 Report

The authors try to analyze the deleted data remaining on the device and the possibility of recovery in order to improve user privacy for smartphones running Android platforms 9 and 10. . Analysis of deleted data is performed based on three data erasure scenarios: data erasure using the app's own function, data erasure using the system app's data and cache wipe function, and uninstalling installed apps. It shows the potential privacy issues of users that may face while using Android 9 and 10 platforms due to loss of the recovered data. It also highlights the improvement in the security of personal user information by erasing the traces of deleted data that remain in the journal area and directory entry area of the file system used on Android 9 and 10 platforms.

--------This is an interesting topic and relevant to its community.-----------

But I have a few notes where the paper can be scientifically improved:

  1. It would be helpful for the reader to state the aim of the paper in the introduction and still the research question the authors want to answer.
  2. The topic is very interesting, it would be useful to define the problem of data erasure and recovery in a "Status of Problems" section for quick understanding. And also to see how these problems were solved by different authors in the literature.
  3. One question: don't the authors present a solution to the problem, or are they only supposed to show and analyze scenarios?
  4. Authors talked about the methods to improve personal data protection for future research. It would be good to list these methods.
  5. Why little literature? There are also many references on the subject

Author Response

(The authors gave the same response as above.)

Reviewer 3 Report

sensors-1676606

Nowadays, data stored in the device may possibly be exposed to various cyber security threats such as personal information leakage and identity theft. Existing research for protection of personal information on the existing Android platform was conducted only on Android platform 6 or lower. In this paper, the authors analyze the deleted data remaining on the device and the possibility of recovery to improve user privacy for smartphones using Android platforms 9 and 10.

This paper is organized well and based on concrete technical operations.

My comments are listed as below:

  1. what are the differences between Android platform 6 or lower and Android platforms 9 and 10 mentioned in this paper? This should be detailed in the context.
  2. please detail the significance of the author’s work.
  3. Since there is no comparison between existing work and the method proposed in the paper, the efficiency and the advantage of the proposed methods remain unknown. The following references may be included for comparison.

[1] M. Petrov, "Android Password Managers and Vault Applications: Data Storage Security Issues Identification," Journal of Information Security and Applications, vol. 67, p. 103152, 2022.

[2] K.-K. R. Choo, Y. Fei, Y. Xiang, and Y. Yu, "Embedded device forensics and security,"  vol. 16, ed: ACM New York, NY, USA, 2016, pp. 1-5.

[3] L. Yang, C. Li, T. Wei, F. Zhang, J. Ma, and N. Xiong, "Vacuum: An Efficient and Assured Deletion Scheme for User Sensitive Data on Mobile Devices," IEEE Internet of Things Journal, 2021.

Some sentences should be rewritten, which include, but are not limited to:

  1. line 181~182
  2. line 14~16

Author Response

(The authors gave the same response as above.)

Round 2

Reviewer 3 Report

The paper in its current form can be published in this journal.